# A protein phosphatase network controls the temporal and spatial dynamics of differentiation commitment in human epidermis

Ajay Mishra[1,2†], Bénédicte Oulès[1†], Angela Oliveira Pisco[1†], Tony Ly[3,4‡], Kifayathullah Liakath-Ali[1‡], Gernot Walko[1], Priyalakshmi Viswanathan[1], Matthieu Tihy[1,5], Jagdeesh Nijjher[1], Sara-Jane Dunn[6,7], Angus I Lamond[3], Fiona M Watt[1]*

[1]Centre for Stem Cells and Regenerative Medicine, King's College London, London, United Kingdom; [2]Department of Chemical Engineering and Biotechnology, Cambridge Infinitus Research Centre, University of Cambridge, Cambridge, United Kingdom; [3]Centre for Gene Regulation and Expression, School of Life Sciences, University of Dundee, Dundee, United Kingdom; [4]Wellcome Centre for Cell Biology, Institute of Cell Biology, School of Biological Sciences, University of Edinburgh, Edinburgh, United Kingdom; [5]Laboratory of Cerebral Physiology, Université Paris Descartes, Paris, France; [6]Microsoft Research, Cambridge, United Kingdom; [7]Wellcome Trust - Medical Research Council Cambridge Stem Cell Institute, University of Cambridge, Cambridge, United Kingdom

*For correspondence: fiona.watt@kcl.ac.uk

[†]These authors contributed equally to this work
[‡]These authors also contributed equally to this work

**Abstract** Epidermal homeostasis depends on a balance between stem cell renewal and terminal differentiation. The transition between the two cell states, termed commitment, is poorly understood. Here, we characterise commitment by integrating transcriptomic and proteomic data from disaggregated primary human keratinocytes held in suspension to induce differentiation. Cell detachment induces several protein phosphatases, five of which - DUSP6, PPTC7, PTPN1, PTPN13 and PPP3CA – promote differentiation by negatively regulating ERK MAPK and positively regulating AP1 transcription factors. Conversely, DUSP10 expression antagonises commitment. The phosphatases form a dynamic network of transient positive and negative interactions that change over time, with DUSP6 predominating at commitment. Boolean network modelling identifies a mandatory switch between two stable states (stem and differentiated) via an unstable (committed) state. Phosphatase expression is also spatially regulated in vivo and in vitro. We conclude that an auto-regulatory phosphatase network maintains epidermal homeostasis by controlling the onset and duration of commitment.
DOI: https://doi.org/10.7554/eLife.27356.001

## Introduction

Commitment is a transient state during which a cell becomes restricted to a particular differentiated fate. Under physiological conditions, commitment is typically irreversible and involves selecting one differentiation pathway at the expense of others (*Nimmo et al., 2015*). While commitment is a well-defined concept in developmental biology, it is still poorly understood in the context of adult tissues (*Simons and Clevers, 2011*; *Semrau and van Oudenaarden, 2015*; *Nimmo et al., 2015*). This is because end-point analysis fails to capture dynamic changes in cell state, and rapid cell state

transitions can depend on post-translational events, such as protein phosphorylation and dephosphorylation (*Avraham and Yarden, 2011*).

We set out to examine commitment in human interfollicular epidermis, which is a multi-layered epithelium formed by keratinocytes and comprises the outer covering of the skin (*Watt, 2014*). The stem cell compartment lies in the basal layer, attached to an underlying basement membrane. Cells that leave the basal layer undergo a process of terminal differentiation as they move through the suprabasal layers. In the final stage of terminal differentiation, the cell nucleus and cytoplasmic organelles are lost and cells assemble an insoluble barrier, called the cornified envelope, which is formed of transglutaminase cross-linked proteins and lipids (*Watt, 2014*). We have previously shown that keratinocytes can commit to terminal differentiation at any phase of the cell cycle, and upon commitment they are refractory to extracellular matrix-mediated inhibition of differentiation (*Adams and Watt, 1989*).

Although there are currently no markers of epidermal commitment, we have previously used suspension-induced differentiation of disaggregated human keratinocytes in methylcellulose-containing medium (*Adams and Watt, 1989*) to define its timing. Here, we use this simple experimental model to identify markers of commitment and discover why it is a transient state.

## Results

### Dynamic expression of protein phosphatases during suspension-induced differentiation

Since keratinocytes increase in size as they differentiate (*Adams and Watt, 1989*), we enriched for undifferentiated cells by filtration prior to placing them in suspension for up to 12 hr (*Figure 1a*). By determining when cells recovered from suspension could no longer resume clonal growth on replating (*Figure 1b*; *Figure 1—figure supplement 1a*), we confirmed that there is a marked drop in colony forming ability between 4 and 8 hr. This correlates with an increase in the proportion of cells expressing the terminal differentiation markers involucrin (IVL) and transglutaminase 1 (TGM1) (*Figure 1c,d*; *Figure 1—figure supplement 1b*) and downregulation of genes that are expressed in the basal layer of the epidermis, including integrin α6 (ITGα6) and TP63 (*Figure 1d*) and the stem cell markers DLL1 and Lrig1 (*Tan et al., 2013*; *Figure 1—figure supplement 1c*). As expected, Ki67 expression was also reduced in suspension, reflecting the drop in proliferation upon differentiation (*Figure 1—figure supplement 1c*).

To define commitment at the molecular level, we next collected keratinocytes after 4, 8 and 12 hr in suspension and performed genome-wide transcriptomics, using Illumina-based microarrays, and proteome-wide peptide analysis, by SILAC-Mass-Spectrometry (MS) (*Figure 1e,f*; *Figure 1—figure supplement 1d–g*; *Supplementary file 1*, *2*). Keratinocytes collected immediately after trypsinisation served as the 0 hr control. When comparing the starting cell population (0 hr) with cells suspended for 4, 8 or 12 hr, t-SNE plots of differentially expressed genes (*Figure 1e*) and unsupervised hierarchical clustering of genes and proteins (*Figure 1f*; *Figure 1—figure supplement 1d*) indicated that the 4 hr samples clustered separately from the 8 and 12 hr samples. GO term enrichment analysis of differentially expressed transcripts or proteins showed enrichment of terms associated with epidermal differentiation at 8 and 12 hr (*Figure 1—figure supplement 1e,g*) (*Sen et al., 2010*; *Mulder et al., 2012*), which is consistent with the drop in clonogenicity seen at these time points.

When we compared the significantly differentially expressed proteins (p-value<0.05) that changed ≥2 fold at one or more time points with their corresponding transcripts (*Figure 1g–i*), there was a moderately positive correlation at 8 and 12 hr (Pearson correlations of 0.51 and 0.68, respectively), consistent with the correlation between bulk mRNA and protein levels seen in previous studies of mammalian cells (*Schwanhäusser et al., 2011*; *Ly et al., 2014*). However, at 4 hr transcripts and proteins were only weakly correlated (Pearson correlation 0.19, *Figure 1g*).

The poor correlation between protein and transcript levels at 4 hr suggested a potential role for post-transcriptional mechanisms in regulating commitment. To investigate this, we performed unbiased SILAC-MS-based phospho-proteomic analysis. SILAC-labelled peptides isolated from cells at 0, 4 and 8 hr time points were enriched for phosphopeptides using HILIC pre-fractionation and titanium dioxide affinity chromatography (*Supplementary file 3*). Over 3500 high confidence phosphorylation sites were identified with an Andromeda search score >= 30 and quantified at both the 4

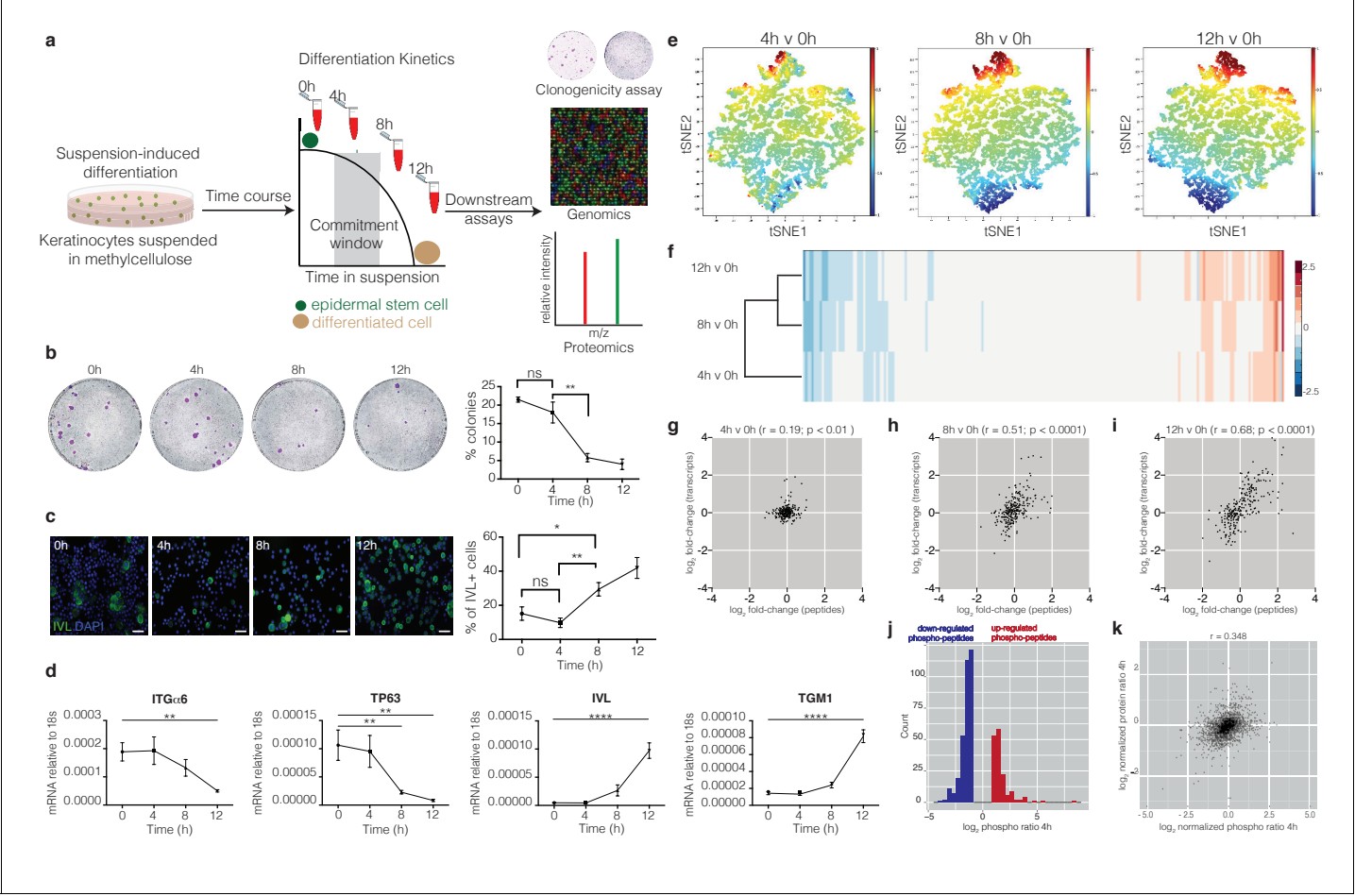

**Figure 1.** Genomic and proteomic analysis identifies protein dephosphorylation events that correlate with commitment. (a) Schematic of experimental design. (b) Colony formation by cells harvested from suspension at different times. Representative dishes are shown together with % colony formation (n = 2 independent experiments, n = 3 dishes per condition per experiment; p-values calculated by Tukey's multiple comparison test). (c) Cells isolated from suspension at different time points were labelled with anti-involucrin (IVL) antibody (green) and DAPI as nuclear counterstain (blue). IVL-positive cells were counted using ImageJ (n = 3 independent cultures; more than 300 cells counted per condition. p-value calculated by two-tailed t-test). Scale bars: 50 μm (d) RT-qPCR quantification of ITGα6, TP63, IVL and TGM1 mRNA levels (relative to 18 s expression) (n = 3 independent cultures). (e) t-SNE plot of genome-wide transcript expression by keratinocytes placed in suspension for different times. The t-SNE algorithm takes a set of points in a high-dimensional space and finds a faithful representation of those points in a lower-dimensional space, typically the 2D plane. (f) Heatmap representing hierarchical clustering of differentially expressed proteins (p<0.05). (g–i) Dot plots correlating expression of significantly differentially expressed peptides (p<0.05) that change twofold relative to 0 hr in at least one of the time points, with their corresponding differentially expressed transcripts. Pearson correlations (r) are indicated. (j) Histogram of normalised SILAC ratios corresponding to high confidence phosphorylation sites that differ between 0 and 4 hr. (k) Scatter plot correlating $\log_2$ normalised SILAC ratios for total protein changes (y-axis) with $\log_2$ phospho-peptide ratios (x-axis) between 0 and 4 hr. (b, c) *p<0.05; **p<0.01; ns = non-significant).

DOI: https://doi.org/10.7554/eLife.27356.002

The following figure supplement is available for figure 1:

**Figure supplement 1.** Clonal growth, genomic and proteomic analysis of suspension-induced terminal differentiation.

DOI: https://doi.org/10.7554/eLife.27356.003

and 8 hr time points. At 4 hr approximately two thirds of the changes involved dephosphorylation (*Figure 1j*), but these dephosphorylation events could not be attributed to decreases in protein abundance, as shown by the discordance between total protein abundance and changes in protein phosphorylation (*Figure 1k*).

We next identified protein phosphatases that were differentially expressed between 0 and 4 hr. We intersected the proteomic and genomic datasets, revealing 47 differentially expressed phosphatases, 22 of which were upregulated (*Figure 2a*). Interrogation of published datasets revealed that

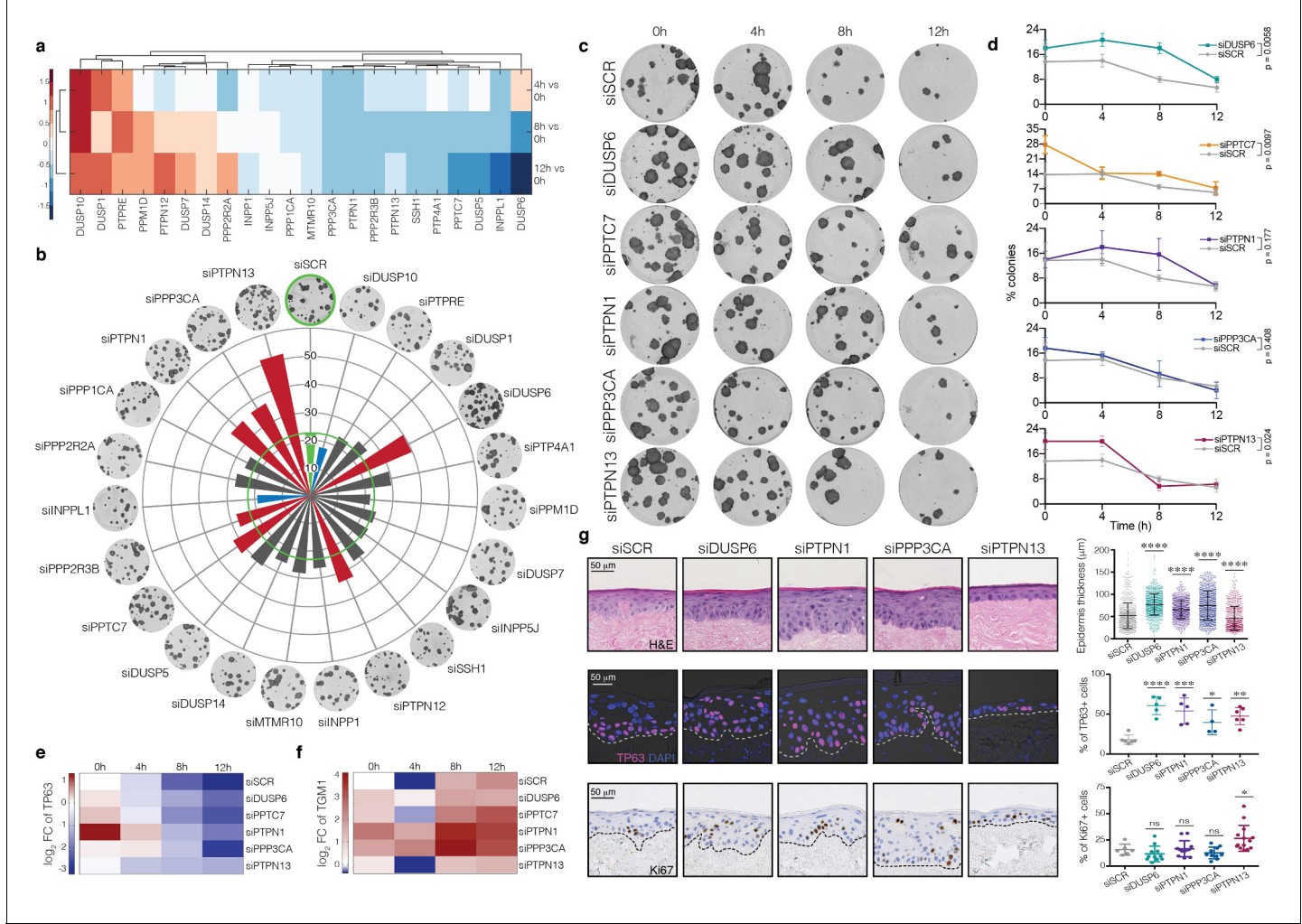

**Figure 2.** Functional screen identifies candidate phosphatases that regulate commitment. (**a**) Heatmap showing differential expression at 4, 8 and 12 hr relative to 0 hr of phosphatases that are upregulated at 4 hr in the microarray dataset. (**b**) Effect of knocking down the 22 phosphatases identified in (**a**) on clonal growth of keratinocytes. Values plotted are average % clonal growth in n = 3 independent screens with n = 3 independent cultures per screen. Green: siSCR control. Red, blue: phosphatases with statistically significant effects on colony formation are shown (red: increase; blue: decrease). Grey: no statistically significant effect. (**c, d**) Effect of knockdowns on clonal growth after 0, 4, 8 or 12 hr in suspension. (**c**) Representative dishes. (**d**) Quantitation of mean % clonal growth ± SD (n = 3 independent samples). p-values generated by unpaired T-test. (**e, f**) RT-qPCR quantification of TP63 (**e**) and TGM1 (**f**) mRNA levels (relative to 18 s expression) in the same conditions as in (**c**). n = 3 independent transfections. (**g**) Epidermal reconstitution assay following knockdown of DUSP6, PTPN1, PPP3CA or PTPN13. n = 2 independent transfections. Top row shows representative H and E images. Epidermal thickness was quantified in multiple fields from eight sections per replicate ± SD relative to scrambled control (siSCR). Middle row shows staining for TP63 (pink) with DAPI nuclear counterstain (blue). % DAPI-labelled nuclei that were TP63+ was quantified in n = 2–3 fields per replicate. Bottom row shows staining for Ki67 (brown) with haematoxylin counterstain (blue). % Ki67+ nuclei was quantified in n = 3–6 fields per replicate. Error bars represent mean ± s.d. p-values were calculated using one-way ANOVA with Dunnett's multiple comparisons test (*p<0.05; **p<0.01; ***p<0.0005; ****p<0.0001; ns = non significant).

DOI: https://doi.org/10.7554/eLife.27356.004

The following figure supplements are available for figure 2:

**Figure supplement 1.** Expression in published datasets of the protein phosphatases identified by suspension-induced differentiation.
DOI: https://doi.org/10.7554/eLife.27356.005
**Figure supplement 2.** Effects of phosphatase knockdown on keratinocyte growth and differentiation.
DOI: https://doi.org/10.7554/eLife.27356.006
**Figure supplement 3.** Effect of shRNA-mediated knockdown of the phosphatases on epidermal reconstitution.
DOI: https://doi.org/10.7554/eLife.27356.007
**Figure supplement 4.** Workflow for automated quantification of the epidermal thickness.
DOI: https://doi.org/10.7554/eLife.27356.008

these phosphatases were also dynamically expressed during calcium-induced stratification of human keratinocytes (*Hopkin et al., 2012*) and differentiation of reconstituted human epidermis (*Lopez-Pajares et al., 2015*) (*Figure 2—figure supplement 1a,b*).

## Identification of protein phosphatases that regulate commitment

To examine the effects on keratinocyte self-renewal of knocking down each of the 22 phosphatases identified in the screen, we transfected primary human keratinocytes with SMART pool siRNAs and measured colony formation in culture (*Figure 2b*; *Figure 2—figure supplement 2a*). Live-cell imaging was used to monitor cell growth for 3 days post-transfection (*Figure 2—figure supplement 2b*). Knocking down seven of the phosphatases significantly increased clonal growth (*Figure 2b*), with five phosphatases – DUSP6, PPTC7, PTPN1, PTPN13 and PPP3CA - having the most pronounced effect (p-value<0.001). Silencing of these five phosphatases also increased colony size and delayed the loss of colony forming ability in suspension (*Figure 2c,d*; *Figure 2—figure supplement 2c–d*). Consistent with these findings, phosphatase knockdown delayed the decline in TP63 and increase in TGM1 levels during suspension-induced differentiation (*Figure 2e,f*; *Supplementary file 4*). Knocking down each phosphatase did not have a major effect on the growth rate of keratinocytes and live-cell imaging did not reveal any apoptosis (*Figure 2—figure supplement 2e*). Cumulatively, the effects on keratinocyte self-renewal and differentiation suggest that DUSP6, PPTC7, PTPN1, PTPN13 and PPP3CA are pro-commitment protein phosphatases.

The effects of knocking down DUSP10 at 0 hr differed from those of the other five phosphatases. DUSP10 knockdown significantly reduced clonogenicity (p-value<0.001; *Figure 2b*; *Figure 2—figure supplement 2a*) and, in contrast to the other phosphatases, decreased the growth rate of keratinocytes (*Figure 2—figure supplement 2b,e*). In addition, whereas expression of DUSP6, PPTC7, PTPN1, PTPN13 and PPP3CA declined by 8 hr in suspension, DUSP10 expression remained high (*Figure 2a*). Taken together, their effects on keratinocyte self-renewal and differentiation suggest that all six are commitment-associated protein phosphatases, with DUSP10 differing from the other phosphatases in potentially antagonising commitment.

To examine the effect of knocking down the pro-commitment phosphatases on the ability of keratinocytes to reconstitute a multi-layered epithelium, we seeded cells on de-epidermised human dermis and cultured them at the air-medium interface for three weeks (*Figure 2g*; *Figure 2—figure supplement 2f,g*). Knockdown of DUSP6, PTPN1, PPP3CA and PTPN13 did not prevent cells from undergoing terminal differentiation, as evidenced by suprabasal expression of involucrin and accumulation of cornified cells (*Figure 2g*; *Figure 2—figure supplement 2f*). However, the number of TP63-positive cells was increased, whether measured as a proportion of the total number of cells (*Figure 2g*) or per length of basement membrane (*Figure 2—figure supplement 2g*). Knockdown of DUSP6, PTPN1 or PPP3CA increased epidermal thickness without increasing the proportion of proliferative, Ki67-positive cells (*Figure 2g*; *Figure 2—figure supplement 2g*). Conversely, PTPN13 knockdown led to an increase in the percentage of Ki67-positive cells and a slight reduction in epidermal thickness (*Figure 2g*), which could reflect an increased rate of transit through the epidermal layers. In addition, Ki67 and TP63-positive cells were no longer confined to the basal cell layer of epidermis reconstituted following phosphatase knockdown but were also present throughout the viable suprabasal layers.

To complement our studies with siRNAs, which achieve transient knockdown, we performed stable knockdown of the pro-commitment phosphatases with shRNA lentiviral vectors (two different shRNAs per target; *Figure 2—figure supplement 3a*) and performed epidermal reconstitution experiments (*Figure 2—figure supplement 3b,c*). We also developed a tool for automated, unbiased measurements of epidermal thickness (*Figure 2—figure supplement 4*, *Source code 1*). No apoptotic cells were observed in reconstituted epidermis, as evaluated by lack of Caspase-3 labelling. Consistent with the effects of siRNA-mediated knockdown, stable knockdown of DUSP6, PPTC7, PTPN1 and PPP3CA increased epidermal thickness, whereas PTPN13 knockdown did not (*Figure 2—figure supplement 3b,c*). We also confirmed expansion of the TP63+ compartment (*Figure 2—figure supplement 3d*). Virtually all Ki67+ cells co-expressed TP63 (*Figure 2—figure supplement 3d,e*). Furthermore, many suprabasal cells co-expressed TP63 and IVL (*Figure 2—figure supplement 3f*). Thus, on both transient and stable knockdown, the transition from the stem to the differentiated cell compartment was disturbed (*Figure 2—figure supplement 3f*).

## Regulation of ERK MAPK and AP1 transcription factors

To identify the signalling networks affected by upregulation of phosphatases during commitment, we performed GO analysis of ranked peptides that were dephosphorylated at 4 hr. The top enriched pathways were ErbB1 signalling, adherens junctions, insulin signalling and MAPK signalling

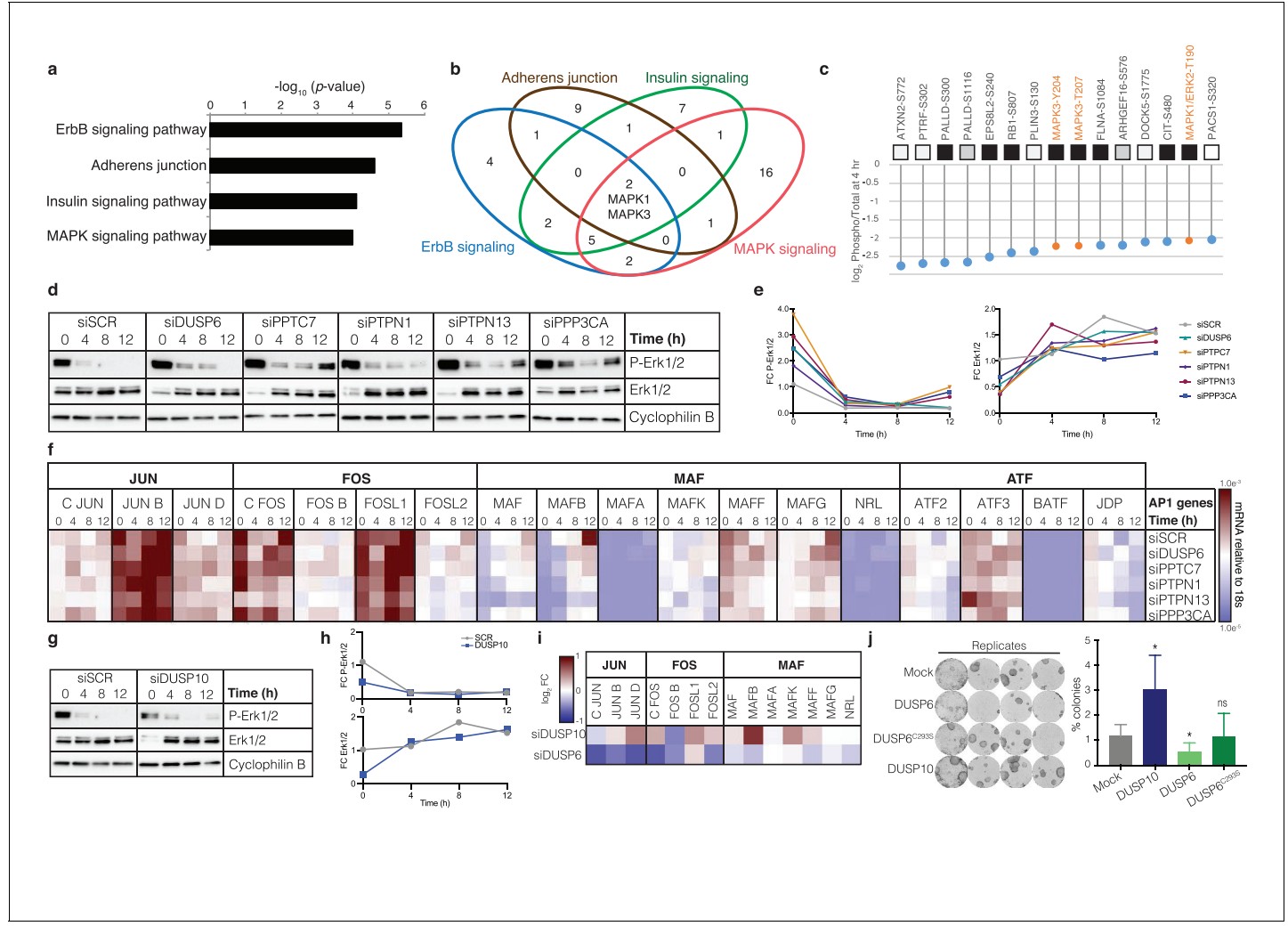

**Figure 3.** Pro-commitment phosphatases regulate MAPK signalling and AP1 transcription factors. (a) Gene Ontology (GO) term enrichment analysis of ranked peptides dephosphorylated at 4 hr. (b) Venn diagram showing intersection of signalling pathways regulated at 4 hr. (c) Top 15 dephosphorylated peptide sites at 4 hr, showing ratio between change in phospho-peptides and change in total protein. Highlighted in orange are phosphorylations on MAPKs. (d, e) Representative western blot (d) and quantification (e) showing phospho-ERK1/2 and total ERK1/2 in cells harvested after 0, 4, 8 and 12 hr in suspension. Cyclophilin B: loading control. (f) Heatmap represents mRNA expression (relative to 18 s RNA) of AP1 transcription factor superfamily members during suspension-induced differentiation post-phosphatase knockdown (n = 3; values plotted are means of three independent transfections). See *Supplementary file 6* for p-values generated by two-way ANOVA. (g–h) Representative western blot, with quantitation, of phospho-ERK1/2 and total ERK1/2 in suspended cells following DUSP10 knockdown. siSCR and loading controls are shown. (i) Heatmap showing mRNA expression (relative to 18 s mRNA) of JUN, FOS and MAF family members after DUSP6 and DUSP10 knockdown (values plotted are means of three independent transfections normalised against scrambled control; see *Supplementary file 7* for p-values generated by two-way ANOVA). (j) Clonal growth (representative dishes and quantification) following doxycycline-induced over-expression of DUSP10, DUSP6 and mutant DUSP6^C293S in primary keratinocytes (n = 3 independent cultures). p-values were calculated using one-way ANOVA with Dunn's multiple comparisons test (*p<0.05; ns = non significant).

DOI: https://doi.org/10.7554/eLife.27356.009

The following figure supplement is available for figure 3:

**Figure supplement 1.** Effects of DUSP10 knockdown and DUSP6 and DUSP10 over-expression.

DOI: https://doi.org/10.7554/eLife.27356.010

(*Figure 3a*). Several of the proteins we identified are components of more than one pathway (*Figure 3b*) and all have been reported previously to regulate epidermal differentiation (*Connelly et al., 2010*; *Haase et al., 2001*; *Kolev et al., 2008*; *Trappmann et al., 2012*; *Scholl et al., 2007*). In particular, constitutive activation of ERK delays suspension-induced differentiation (*Haase et al., 2001*).

We next ranked protein phosphorylation sites according to the $\log_2$-fold decrease at 4 hr, plotting the ratio between the change in phosphorylation sites and the change in total protein (*Supplementary file 5*). To specifically identify dephosphorylation events, we excluded from the ranking phosphorylations that remained constant while total protien abundance increased by more than 0.5 in $\log_2$. Consistent with the predicted dynamic interactions between signalling pathways (*Figure 3b*), phosphorylation sites on MAPK1 (ERK2) and MAPK3 (ERK1) were identified in the top 15 most decreased sites (*Figure 3c*). Other proteins in the top 15 included components or regulators of the cytoskeleton (FLNA), Rho signalling (DOCK5, ARHGEF16, CIT) and EGFR signalling (EPS8), again consistent with the GO terminology analysis.

We performed western blotting to confirm that ERK1/2 activity was indeed modulated by suspension-induced terminal differentiation and by the candidate pro-commitment phosphatases (*Figure 3d,e*). As reported previously, the level of phosphorylated ERK1/2 diminished with time in suspension (*Janes et al., 2009*) (*Figure 3d*). siRNA-mediated knockdown of DUSP6, PPTC7, PTPN1, PTPN13 or PPP3CA resulted in higher levels of phosphorylated ERK1/2 relative to the scrambled control, both at 0 hr and at some later time points (*Figure 3d,e*). These effects are consistent with the known requirement for ERK MAPK activity to maintain keratinocytes in the stem cell compartment (*Trappmann et al., 2012*).

Transcriptional regulation of epidermal differentiation is mediated by the Activator Protein 1 (AP1) family of transcription factors (*Eckert et al., 2013*), which is the main effector of the MAPK and ErbB signalling cascades (*Chang and Karin, 2001*). Quantification of the levels of AP1 transcripts during suspension-induced terminal differentiation revealed that different AP1 factors changed with different kinetics, as reported previously (*Gandarillas and Watt, 1995*) (*Figure 3f*). Notably, the level of several members of the MAF subfamily of AP1 factors (MAF, MAFB and MAFG) significantly increased during differentiation, consistent with recent evidence that they mediate the terminal differentiation program in human keratinocytes (*Lopez-Pajares et al., 2015*). In line with these observations, knockdown of individual pro-commitment phosphatases reduced the induction of MAF AP1 factors in suspension (*Figure 3f*; *Supplementary file 6*). These experiments are consistent with a model whereby induction of phosphatases in committed keratinocytes causes dephosphorylation of ERK MAPK and prevents the increase in expression of AP1 transcription factors that execute the terminal differentiation program.

Given that DUSP10 knockdown differed from the other protein phosphatases in decreasing the clonogenicity (*Figure 2b*; *Figure 2—figure supplement 2a*) and growth rate of keratinocytes (*Figure 2—figure supplement 2b,e*), we examined the effect of DUSP10 knockdown on ERK1/2 activity and AP1 factor expression. Unlike knockdown of DUSP6, PPTC7, PTPN1, PTPN13 or PPP3CA, knockdown of DUSP10 (*Figure 3—figure supplement 1a,b*) did not increase ERK1/2 activity (*Figure 3g,h*). Consistent with its known regulation of p38 MAPK (*Caunt and Keyse, 2013*), DUSP10 knockdown increased phospho-p38 at 0 hr; however, this effect was not observed at later times in suspension (*Figure 3—figure supplement 1c,d*). When the effects of DUSP10 knockdown on AP1 transcription factors were compared with those of DUSP6, it was clear that these phosphatases differentially regulated several AP1 factors, including members of the JUN, FOS and MAF subfamilies (*Figure 3i*; *Supplementary file 7*).

We confirmed the differing effects of DUSP10 and DUSP6 on cell fate by using either Cumate or Doxycycline inducible overexpression in human keratinocytes (*Figure 3j*; *Figure 3—figure supplement 1e–g*). Whereas overexpression of DUSP10 increased colony formation, overexpression of DUSP6 reduced clonal growth (*Figure 3j*). A dominant negative mutant of DUSP6 (C293S) lacking phosphatase activity (*Okudela et al., 2009*) had no effect (*Figure 3j*; *Figure 3—figure supplement 1f,g*).

# A protein phosphatase interaction network acts as a switch to transition cells between the stem and differentiated cell compartments

We next examined whether the six phosphatases we identified experimentally could form a single phosphatase-phosphatase interaction network governing commitment and differentiation. To do so, we employed a Boolean network abstraction and constructed a model in which all phosphatases, DUSP6, PPTC7, PTPN1, PTPN13, PPP3CA and DUSP10, were allowed to interact either positively or negatively (*Figure 4—figure supplement 1a*). For this we utilised the concept of an Abstract Boolean Network (ABN) (*Yordanov et al., 2016*), which by assigning definite or possible interactions, implicitly defines approximately $10^{17}$ concrete different Boolean network models (*Figure 4—figure supplement 1a*). In accordance with this modelling formalism, we abstracted discretised gene expression levels as 'on' or 'off' if their mean expression was respectively higher or similar/lower than the average of all genes (*Supplementary file 8*). From the set of all possible Boolean networks (the ABN), we sought to test whether there would be a network consistent with the discretised gene expression states over the differentiation time course. To this end, we employed the automated reasoning approach encapsulated in the tool RE:IN, as described by *Dunn et al., 2014*. We encoded a set of experimental constraints comprising the different time points analysed during suspension-induced differentiation (*Figure 1a*). These constraints define whether each gene should be on or off at each time step along the differentiation trajectory (Materials and methods). Using this approach, we formally deduced that a single Boolean network was unable to recapitulate the measured gene expression dynamics (*Figure 4—figure supplement 1a*).

As we were not able to find a single interaction network sufficient to model the discretised gene expression states, we further investigated each individual time point to identify whether the network topology might differ over the time course in suspension. To obtain the necessary experimental data to identify critical network interactions, we performed individual knockdowns of the phosphatases after 0, 4, 8 or 12 hr in suspension. Next, we measured by RT-qPCR the effect that each individual knockdown had on the mRNA level of any of the other phosphatases (*Figure 4—figure supplement 2a*) at each time point. We used the fold-change of the interaction to inform its directionality (activating or inhibitory) only if it was statistically significant (p-value<0.05, *Supplementary file 9*). This led to the inferred mechanistic networks depicted in *Figure 4a*, where the node colours show fold-change in each phosphatase with time in suspension, relative to the 0 hr time-point. Arrows indicate positive effects on expression and T-bars show inhibitory effects. These networks reveal different interactions at play at each of the time points, suggesting that the network topology reconfigures during commitment and differentiation.

In light of these inferred mechanistic interactions, we again employed a Boolean network abstraction to explore the dynamics of epidermal stem cell commitment. Using the discretised gene expression levels defined above (*Supplementary file 8*), we sought to test whether the network topologies derived for each time point (*Figure 4a,b*) were consistent with the data. For this we used an extension of RE:IN, called RE:SIN, which allows the user to explore switching networks, as described by *Shavit et al. (2016)*. RE:SIN allowed us to determine whether the inferred networks could recapitulate the dynamic changes in gene expression by similarly encoding these as expected states on the commitment trajectory (*Figure 4—figure supplement 1b*). We first found that the networks inferred by the genetic knockdown experiments could not satisfy the experimental constraints. This suggested that there were additional interactions between the phosphatases. To identify such interactions, we analysed the effect that each individual knockdown had on the protein levels of the other phosphatases (*Figure 4—figure supplement 2d*; *Supplementary file 10*) and the effect of DUSP6 and DUSP10 overexpression on phosphatase mRNA levels (*Figure 4—figure supplement 2e*). We included those as possible interactions, again using the concept of an ABN at each time step (*Figure 4—figure supplement 1c*; *Supplementary file 8*).

By defining an ABN at each time point that incorporates the possible phosphatase interactions deduced from the knockdown experiments, we found that the experimental constraints could be satisfied. Furthermore, our analysis revealed which of the possible interactions were required to meet these constraints. These could therefore be considered as definite interactions (*Figure 4c*, compare with *Figure 4—figure supplement 1c*). We thus used the approach encapsulated in RE:SIN to uncover the networks governing differentiation at each time point.

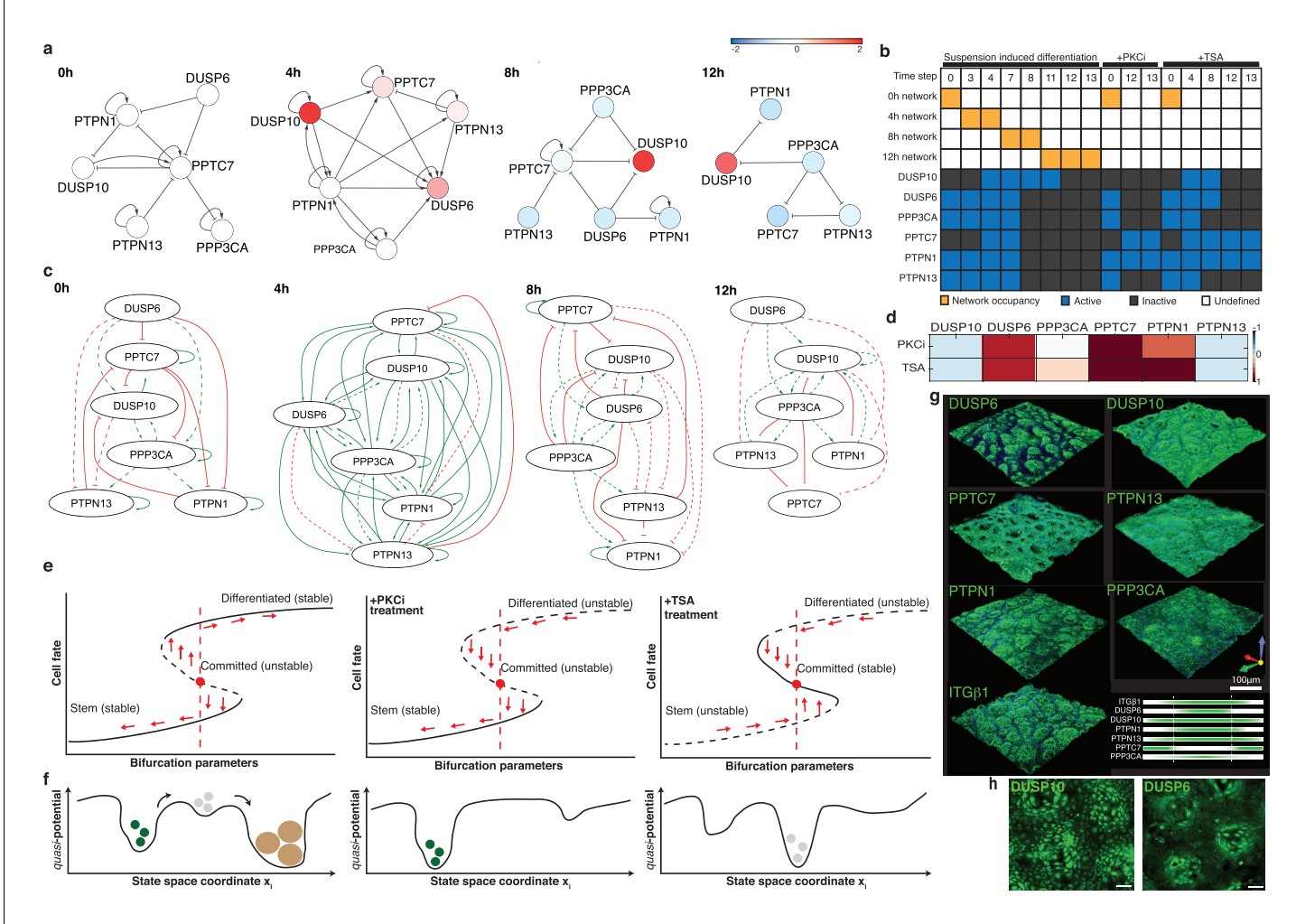

**Figure 4.** An autoregulatory network of phosphatases controls commitment. (a) Colours represent log2 fold-change in phosphatase mRNA expression relative to 0 hr (values plotted are means of three independent experiments normalised against siSCR control). (b) Experimental observations that are encoded as constraints on the Boolean network trajectories. We defined three experimental constraints (gene expression changes during suspension-induced differentiation in the absence of pharmacological inhibitors, as well as under TSA and PKCi treatments). Each constraint encodes discrete gene expression states at the indicated time steps. For cells in the absence of drugs we imposed a switching scheme, whereby the system must change the representative network in order to achieve the expression constraints. Yellow boxes indicate the network that represents the system at that step; blank boxes for a given step (column) indicate that we did not impose a specific network, so the system can remain in the current network or switch forwards. Active means the gene is considered to be 'on' at that step, inactive means to be 'off' at that step. The discretisation is available in **Supplementary file 9**. (c) Networks able to satisfy the model constraints of the Boolean formalism in (b) are depicted. Solid lines show interactions already calculated in (a), while dashed lines are possible interactions inferred from **Figure 4—figure supplement 1**. See also **Supplementary file 8** and **9**. (d) Heatmap represents mRNA expression (relative to 18 s mRNA) of individual phosphatases in cells treated in suspension for 12 hr with PKCi or TSA (values plotted are the means of three independent experiments normalised against vehicle-treated control). (e, f) Representation of commitment as two saddle-node bifurcations in a direction $x_i$ of the state space for control cells or cells treated with PKCi or TSA. In the control both stem and differentiated cell states are stable (attractors), while commitment is an unstable state. Since the 0 hr network is able to reach the expression constraint for PKCi at 12 hr, we hypothesise that on PKCi treatment, the only stable state is the stem state. On TSA treatment there is a mandatory switch from the 0 hr network but the 12 hr network cannot be reached at any time point; we therefore hypothesise that commitment becomes a stable state while the stem and differentiated cell states are unstable. (g) 3D-volume rendered confocal images of wholemounts of human epidermis labelled with antibodies against commitment phosphatases or ITGβ1 (green) and counterstained with DAPI (blue). The distribution of each phosphatase relative to ITGβ1 is also shown graphically. (h) 3D-volume rendered confocal images of primary keratinocytes cultured on PDMS topographic substrates and labelled with antibodies to DUSP10 and DUSP6. Scale bars: 100 μm.

DOI: https://doi.org/10.7554/eLife.27356.011

The following figure supplements are available for figure 4:

**Figure supplement 1.** Simulated Boolean networks.

*Figure 4 continued on next page*

*Figure 4 continued*

DOI: https://doi.org/10.7554/eLife.27356.012

**Figure supplement 2.** Effects on phosphatase expression of knockdowns and treatment with TSA or PKCi.

DOI: https://doi.org/10.7554/eLife.27356.013

**Figure supplement 3.** DUSP6 and DUSP10 distribution in human epidermal wholemounts.

DOI: https://doi.org/10.7554/eLife.27356.014

The Boolean network analysis indicates a key role for DUSP6 at commitment (4 hr), when DUSP6 expression is positively regulated by all the other phosphatases in the network and by a self-activating loop (*Figure 4a*). In contrast, at all other time points the interactions between individual phosphatases were predominantly negative. Several phosphatases, including PTPN1 and PTPN13, showed positive autoregulation at one or more time point. It is also notable that DUSP10 was predominantly negatively regulated by other phosphatases, except at 4 hr, when it was positively regulated by PTPN1 and by an autoregulatory loop (*Figure 4a*).

To test the robustness of the network, we examined the effects of treating keratinocytes with the histone deacetylase inhibitor Trichostatin A (TSA) and a Protein Kinase C inhibitor (PKCi). Both drugs blocked suspension-induced terminal differentiation, as measured by expression of IVL and TGM1, and prevented downregulation of ITGα6 (*Figure 4—figure supplement 2b,c*). However, cells treated with TSA still underwent commitment, as evaluated by loss of colony-forming ability and downregulation of TP63, whereas those treated with the PKC inhibitor did not. At 12 hr in suspension, both drugs reduced expression of DUSP10 and PTPN13 and increased expression of DUSP6 and PPTC7 relative to untreated cells; however, they differentially affected PPP3CA and PTPN1 (*Figure 4d*).

We next used our modelling approach to test whether network switching still occurs under PKCi and TSA treatment using the constrained ABNs that we derived for each stage. As before, we discretised gene expression levels from these experiments as 'on' or 'off' (*Supplementary file 8*). Under both treatments, we found that we could not transit through the networks while respecting the expected expression states, corroborating the experimental findings that terminal differentiation is blocked in these conditions. The PKCi phosphatase expression pattern at 12 hr was compatible with the phosphatase interaction network derived at 0 hr, supporting the conclusion that PKCi arrests cells in the stem cell compartment. In TSA-treated cells the network must switch from that derived at 0 hr to that derived at 4 hr, and subsequently to that derived at 8 hr, or straight from 0 hr to 8 hr. Neither PKCi nor TSA treatment resulted in an expression pattern compatible with the network derived at 12 hr for untreated cells, consistent with the inhibition of differentiation.

The above findings can be summarised using dynamical systems terminology. Epidermal differentiation can be described by two saddle-node bifurcations (*Figure 4e,f*). We start with a single minimum that corresponds to a stable state: the stem cell. Then we pass through a first saddle-node bifurcation where we have two minima, so the system can switch from one state to the other and this corresponds to commitment. Next the global minimum changes and we finally have a second saddle-node bifurcation that leads again to a single steady state, which corresponds to the differentiated cells (*Zhang et al., 2012*). From here, the stem cell state and the terminally differentiated state emerge as stable states, while commitment is inherently unstable and serves as a biological switch.

## Protein phosphatase expression is patterned within human epidermis and responds to topographical cues

Given the role of the protein phosphatases in regulating commitment of keratinocytes in suspension, we examined whether phosphatase expression was also subject to spatial regulation, since spatiotemporal coordination of stem cell behaviour contributes to epidermal homeostasis (*Doupé and Jones, 2012*). By wholemount labelling the basal layer of sheets of human epidermis (*Jensen et al., 1999*), we found that DUSP6, PTPN1 and PPP3CA were most highly expressed in cells with the highest levels of β1 integrin or α6 integrin, which correspond to stem cell clusters (*Jensen et al., 1999*) (*Figure 4g*; *Figure 4—figure supplement 3a,b*). In contrast, PPTC7 was enriched in the integrin-low regions, while PTPN13 and DUSP10 were more uniformly expressed throughout the basal layer (*Figure 4g*; *Figure 4—figure supplement 3a,b*). The inverse relationship between the patterns of

PPP3CA and PPTC7 expression is in good agreement with the network analysis demonstrating negative regulation of PPP3CA by PPTC7 in undifferentiated keratinocytes (0 hr; *Figure 4a*).

The patterned distribution of phosphatases could be recreated in vitro by culturing keratinocytes on collagen-coated PDMS elastomer substrates that mimic the topographical features of the human epidermal-dermal interface (*Viswanathan et al., 2016*). We observed clusters of cells with high levels of DUPS6 on the tops of the features, where stem cells expressing high levels of β1 integrins accumulate (*Viswanathan et al., 2016*) (*Figure 4h*). In contrast, DUSP10 was uniformly expressed regardless of cell position (*Figure 4h*). These results indicate that the phosphatases are subject to spatial regulation that is independent of signals from cells in the underlying dermis.

## Discussion

We have presented evidence that epidermal commitment is a biological switch controlled by a network of protein phosphatases that are regulated spatially and temporally. Applying Boolean network analysis to our experimental data, we were unable to recapitulate the observed changes in phosphatase gene expression using a single ABN, but we could do so by implementing a scheme of network switching. This led us to conclude that the interactions among the commitment phosphatases change in the course of differentiation. The negative feedback loops predominating at 0, 8 and 12 hr are known to result in stable phenotypes because the network is able to counteract additional inputs (*Zeigler et al., 2000*). However, at 4 hr all but one of the interactions were positive. Positive feedback loops lead to instability, because the network amplifies any inputs it receives. This supports the concept of commitment as an unstable state, which aligns with the experimental evidence of that happening within a defined temporal window.

The key role of DUPS6 at commitment fits well with its upregulation by Serum Response Factor, which is known to control keratinocyte differentiation (*Connelly et al., 2010*), and its importance in controlling the activation kinetics and dose-response behaviour of ERK MAPK signalling (*Blüthgen et al., 2009*). However, the involvement of multiple phosphatases in commitment may protect cells from undergoing premature terminal differentiation. Furthermore, their patterned distribution within the epidermal basal layer would be consistent with different phosphatases regulating commitment in different positions along the basement membrane, and with the ability of different external stimuli to trigger differentiation via different intracellular pathways (*Watt, 2016*).

The upregulation of basal layer markers in the suprabasal epidermal layers on knockdown of pro-commitment phosphatases mimics features of psoriatic lesions in which ERK is known to be upregulated (*Haase et al., 2001*). In particular, in the epidermal reconstitution assays there was suprabasal expression of Ki67 and TP63, with TP63+ cells co-expressing IVL. These observations lead us to speculate that commitment is abnormally stabilised in psoriasis, as observed in TSA treated keratinocytes in suspension. Given that protein phosphatases within the commitment network are known to regulate expression of inflammatory cytokines (*Caunt and Keyse, 2013*), deregulated phosphatase expression could potentially contribute to the pathophysiology of psoriasis.

In conclusion, the simple experimental model of suspension-induced keratinocyte differentiation has enabled us to capture some of the complex transcriptional and post-translational events that control the transition from the stem to the differentiated cell compartment. We now have the opportunity to explore, systematically, the key upstream and downstream regulators of protein phosphatases during epidermal homeostasis, injury and disease.

## Materials and methods

### Cell culture

Primary human keratinocytes (strain km) were isolated from neonatal foreskin and cultured on mitotically inactivated 3T3-J2 cells in complete FAD medium, containing one part Ham's F12, three parts Dulbecco's modified eagle medium (DMEM), $10^{-4}$ M adenine, 10% (v/v) FBS, 0.5 µg/ml hydrocortisone, 5 µg/ml insulin, $10^{-10}$ M cholera toxin and 10 ng/ml EGF, as described previously (*Gandarillas and Watt, 1995*). Prior to suspension-induced differentiation in methylcellulose (*Adams and Watt, 1989*) we enriched for stem cells by filtering the disaggregated keratinocytes twice through a 40-µm sterile membrane. For knockdown or overexpression experiments,

keratinocytes were grown in Keratinocyte-SFM medium (Gibco) supplemented with 0.15 ng/ml EGF and 30 µg/ml BPE. All isoforms of PKC were inhibited using 5 µM GF 109203X (Tocris; at lower concentrations GF190203X preferentially inhibits the -α and -β isoforms) and histone deacetylase was inhibited with TSA (Sigma Aldrich, UK). PDMS substrates that mimic the topography of the epidermal-dermal junction were generated as described previously (*Viswanathan et al., 2016*).

## Genome-wide expression profiling

Genome-wide expression profiling was performed using the Illumina BeadArray platform and standard protocols. Data were processed using R (http://www.r-project.org/) or Genespring GX13.1 software. The gene expression data are deposited in the GEO databank (GSE73147).

## Computational analysis of gene expression datasets

Microarray initial processing and normalisation were performed with BeadStudio software. BeadChip internal *p*-values (technical bead replicates) were used to identify genes significantly expressed above the background noise. To filter out genes with signals that were not significant, a p-value of 0.05 was used as the cut-off value and only genes with a p-value<0.05 in at least one sample passed the filter. From the original set of 34,685 gene targets, 23,356 targets met this criterion. The data were imported into GeneSpring v13, normalised using quantile normalisation and the biological replicates averaged for subsequent analysis. We performed pairwise comparison between 0 hr and 4, 8 and 12 hr. Genes showing a fold change higher than two compared to control (and both *p*-values were significant) were subjected to GO analysis with GeneSpring v13 (152 genes between 4 hr and 0 hr, 553 between 8 and 0 hr, 1136 genes between 12 and 0 hr). Hierarchical clustering based on Pearson's uncentered distance was performed on time course gene expression data and the results presented as a heatmap. Additionally, t-SNE plots were generated for the transcriptomics data using the t-SNE package for R.

## Analysis of mechanistic networks

A two-way ANOVA with multiple comparisons corrected using the Holm-Sidak test was used to identify the statistically significant effects of single phosphatase knock-down on the expression of the other phosphatases ($p_{ij}$, *Supplementary file 9*). The weight of each edge was calculated as the inverse p-value for the respective interaction ($w_{ij} = 1/p_{ij}$). For simplicity, we kept only the significant links (p-value<0.05). The networks depicted in *Figure 4a* were drawn with Cytoscape (cytoscape. org, V3.2.1).

## Dynamic Boolean network analysis

The Boolean network analysis was performed using RE:IN and RE:SIN software, which is designed to encode and test whether a single or set of Boolean networks are consistent with observed changes in gene expression states. The methodology has been described previously (*Dunn et al., 2014*; *Yordanov et al., 2016*; *Shavit et al., 2016*). Briefly, a set of possible and definite interactions comprises an Abstract Boolean Network (ABN), which implicitly defines a set of concrete Boolean networks. Each concrete network has a unique topology defined by the different combinations of the possible interactions, which can be present or absent. This allows the user to explore alternative model topologies, and to test the requirement for different interactions that are suggested by experimental data. The software allows the user to seek the set of concrete Boolean network models that are consistent with a set of experimental constraints, and then to use this constrained set of models to make predictions of untested behaviour. Experimental constraints are defined as expected states along network trajectories (*Figure 4b*), which are constructed from discretised gene expression patterns. The software is freely available to use and can be accessed together with tutorials and FAQ at research.microsoft.com/rein.

We considered Boolean networks with synchronous updates, such that each gene in the network updates at each step in the network trajectory. Logical update functions for each gene, which define how the gene switches on or off in response to its regulators, were as defined by *Yordanov et al., 2016*. We first used RE:IN to test whether a single network (one that does not switch along the differentiation trajectory) is consistent with the observed changes in gene expression. We tested all possible single networks by allowing possible positive and negative interactions between each pair

of genes. None of these networks was found to be consistent with the gene expression changes. We next constructed a switching problem in which we defined four ABNs, which individually correspond to the time points measured in our knockdown experiments. We defined the allowable network switching to be monotonic: networks can progress forward in time (i.e. 0 hr to 4 hr, 0 hr to 8 hr etc.) but not backwards (i.e. 8 hr cannot switch to 4 hr or 0 hr). We then used RE:SIN to test whether it was possible to find a sequence of networks that allowed the gene expression changes to be satisfied. We found that this was possible, and, furthermore, identified interactions that were required/disallowed to meet the constraints (*Figure 4c*).

## Defining constraints for Boolean network analysis

The network governing differentiation must be consistent with observed changes in gene expression patterns over time. To investigate whether a given network could recapitulate the observed behaviour, we defined the expected states along a differentiation trajectory starting from the initial gene expression state at 0 hr. Three experimental constraints were defined, corresponding to measured gene expression changes at specific time points during suspension-induced differentiation, PKC inhibition and TSA treatment (*Figure 4b*). Individual gene expression measurements were discretised as 'on' or 'off' if their mean expression was respectively higher or similar/lower than the average of all genes. We set one step in the network trajectory to correspond to 1 hr in the time course, such that we defined expected gene expression states at 0, 4, 8 and 12 hr.

## Generation and analysis of SILAC LC-MS/MS datasets

A pre-confluent keratinocyte culture was split into two separate cultures with FAD$^{-lysine-arginine}$ medium (Sigma) differentially supplemented either with $K^0R^0$ or with $K^8R^{10}$ (stable isotopes of amino acids Lysine and Arginine; Cambridge Isotope Laboratories) (*Ly et al., 2014*). FCS used for SILAC medium was also depleted of Lysine and Arginine (Sigma). Cells were grown in SILAC medium for 5–6 days to reach 70–80% confluence and later harvested for downstream assays. Light labelled cells served as the 0 hr sample whereas heavy labelled cells were suspended in methylcellulose and harvested at 4, 8 and 12 hr. Cell extracts prepared from 0 h cells were mixed individually with 4, 8 and 12 hr samples at a 1:1 ratio of total protein. The mixed samples were then subjected to mass spectrometry (MS).

For measurement of protein abundance changes by SILAC, lysates containing ~80 µg protein were prepared in LDS sample buffer containing reducing agent (TCEP). Proteins were separated by electrophoresis on a 4–12% Bis-Tris NuPAGE gel, which was Coomassie-stained and cut into eight equally sized gel slices. Gel-embedded proteins were reduced with TCEP, alkylated with iodoacetamide, and trypsin-digested to release peptides. Peptides were extracted from gels using 50% acetonitrile (ACN) containing 5% formic acid (FA), dried and resuspended in 5% FA. Peptides were then analysed by LC-MS/MS on a Dionex RSLCnano coupled to a Q-Exactive Orbitrap classic instrument. Specifically, peptides were loaded onto a PepMap100 75 µm x 2 cm trap column, which was then brought in-line with a 75 µm x 50 cm PepMap-C18 column and eluted using a linear gradient over 220 min at a constant flow rate of 200 nl/min. The gradient composition was 5% to 35% B, where solvent A = 2% ACN+0.1% FA and solvent B = 80% ACN+0.1% FA. Peptides were eluted with a linear elution gradient (5% B to 35% B) over 220 min with a constant flow rate of 200 nl/min. An initial MS scan of 70,000 resolution was acquired, followed by data-dependent MS/MS by HCD on the top 10 most intense ions of ≥2+ charge state at 17,500 resolution.

For phosphoenrichment analysis, lysates containing ~2 mg of protein were chloroform:methanol-precipitated. Pellets were resuspended in 8 M urea in digest buffer (0.1 M Tris pH 8 + 1 mM CaCl$_2$), diluted to 4 M urea with additional digest buffer, and digested with Lys-C for 4 hr at 37°C. The digests were diluted to 1 M urea and trypsin-digested overnight at 37°C. Digests were then acidified and desalted using SepPak-C18 vacuum cartridges. Desalted peptides were resuspended in mobile phase A for HILIC (80% ACN + 0.1% TFA). HILIC chromatography was performed using a Dionex Ultimate 3000 with a TSK Biosciences Amide-80 column (250 × 4.6 mm) (*Di Palma et al., 2013*; *McNulty and Annan, 2008*; *Navarro et al., 2011*). Peptides were eluted using an exponential gradient (80% B to 60% B) composed of A (above) and B (0.1% TFA) at 0.4 ml/min over 60 min. 16 fractions were collected from 25 to 60 min. These fractions are enriched for hydrophilic peptides,

including phosphopeptides. The fractions were dried before further phosphoenrichment by titanium dioxide ($TiO_2$).

For $TiO_2$ enrichment, HILIC fractions were resuspended in loading buffer (70% ACN + 0.3% lactic acid + 3% TFA). 1.25 mg of $TiO_2$ (GL Sciences) was added to each fraction and incubated for 10 min to bind phosphopeptides. Beads were washed with loading buffer, and two wash buffers, composed of (1) 70% ACN + 3% TFA and (2) 20% ACN + 0.5% TFA. Phosphopeptides were eluted in two steps, first with 4% of ammonium hydroxide solution (28% w/w $NH_3$) in water for 1 hr and again with 2.6% of ammonium hydroxide solution in 50% ACN overnight. Elutions were collected, dried, resuspended in 5% FA and analysed by LC-MS/MS. The LC-MS analysis was performed similarly to above with the following modifications: peptides were chromatographed on an EasySpray PepMap 75 µm x 50 cm column, and a 'Top30' method was used where the top 30 most intense ions were chosen for MS/MS fragmentation. Raw data were then processed using MaxQuant, which implements the Andromeda search engine, for peptide and protein identification and SILAC quantitation.

For the proteomics dataset, 1155 out of 2024 proteins had a significant p-value<0.05 (calculated in MaxQuant). To identify differentially expressed proteins between time 0 and 4, 8 and 12 hr, we collected per time point the proteins whose absolute ratios were >0.5. For each time point, the proteins were then split according to whether the ratio was positive or negative. Six lists were annotated for GO terminology using GeneSpring. The extensive list of GO-terms was submitted to Revigo (*Supek et al., 2011*) to reduce complexity and the resulting GO-categories depicted in *Figure 1—figure supplement 1g Figure 1f*. The mass spectrometry proteomics data have been deposited with the ProteomeXchange Consortium via the PRIDE partner repository with the dataset identifier PXD003281.

## siRNA nucleofection

siRNA nucleofection was performed with the Amaxa 16-well shuttle system (Lonza). Pre-confluent keratinocytes were disaggregated and resuspended in cell line buffer SF. For each 20 µl transfection (program FF-113), $2 \times 10^5$ cells were mixed with 1–2 µM siRNA duplexes as described previously (*Mulder et al., 2012*). Transfected cells were incubated at ambient temperature for 5–10 min and subsequently replated in pre-warmed Keratinocyte-SFM medium until required for the downstream assay. SMART pool ON-TARGET plus siRNAs (Ambion/GE Healthcare) were used for gene knockdowns. Each SMART pool was a mix of four sets of RNAi oligos. The sequences of the siRNA oligos are provided in the *Supplementary file 11*.

## shRNA infection

shRNA lentiviral vectors were purchased from Sigma-Aldrich (Mission shRNA). Two shRNAs per target were chosen. Their sequences are listed in *Supplementary file 12*. Primary keratinocytes were transduced with lentiviral particles for 48 hr, before being subjected to 2 µg/ml Puromycin selection for 72 hr. Cells were then expanded for experimental analysis.

## Doxycycline and cumate inducible overexpression

For the Doxycycline inducible system we used the pCW57-GFP-2A-MCS lentiviral vector (gift from Adam Karpf; Addgene plasmid # 71783). Primary keratinocytes were transduced with lentiviral particles containing protein expression vector encoding genes for wild-type DUSP6, mutant DUSP6$^{C293S}$ and DUSP10. 2 days post-transduction, cells were subjected to 2 µg/ml Puromycin selection for 3 days and then 1 µg/ml Doxycycline was added to the growth medium.

For Cumate induction, we used the lentiviral QM812B-1 expression vector (System Biosciences). Cells were transduced and selected as described for the Doxycycline system and protein expression was induced by addition of 30 µg/ml Cumate solution to the growth medium.

## Clonogenicity assays

One-hundred, 500 or 1000 keratinocytes were plated on a 3T3 feeder layer per 10 cm diameter dish or per well of a six-well dish. After 12 days, feeders were removed and keratinocyte colonies were fixed in 10% formalin (Sigma) for 10 min then stained with 1% Rhodanile Blue (1:1 mixture of Rhodamine B and Nile Blue A [Acros Organics]). Colonies were scored using ImageJ and clonogenicity was calculated as the percentage of plated cells that formed colonies.

## Skin reconstitution assays

Pre-confluent keratinocyte cultures (km) were disaggregated and transfected with SMART pool siRNAs or non-targeting control siRNAs, or with lentiviral shRNAs. 24 hr post-siRNA transfection or >7 days post-shRNA infection, keratinocytes were collected and reseeded on irradiated de-epidermised human dermis (*Sen et al., 2010*) in six-well Transwell plates with feeders and cultured at the air–liquid interface for 3 weeks. The cultures were then fixed in 10% neutral buffered formalin (overnight), paraffin embedded and sectioned for histological staining. 6 µm thick sections were labelled with haematoxylin and eosin or appropriate antibodies. H and E stained images were acquired with a Hamamatsu slide scanner and analysed using NanoZoomer software (Hamamatsu).

## Automated analysis of skin reconstitution assays

H and E stained sections were scanned with a Hamamatsu slide scanner and analysed in Python. Briefly, tissue sections were automatically detected based on k-means segmentation, and colours were rebalanced. The epidermis was then isolated by colour separation (extraction of haematoxylin color channels by a deconvolution algorithm) (*Ruifrok and Johnston, 2001*). After performing an automatic thresholding, epidermal area was measured by pixel counting. To measure epidermal thickness, the area was divided by a 1-pixel wide reduced-topology (*van der Walt et al., 2014*). Full script is available as Supplementary Source Code 1.

## Immunofluorescence staining of cultured cells and reconstituted skin

For IVL staining of cells in suspension, cells were recovered from methylcellulose, spun down onto a slide using a Cytospin and then fixed with 4% paraformaldehyde and permeabilised. Sections of paraffin embedded reconstituted skin were dewaxed, rehydrated and subjected to heat-mediated antigen retrieval. Immunofluorescence labelling of cells and sections was performed with the appropriate antibodies. Wherever indicated, DAPI was used to label nuclei. Image acquisition was performed using a Nikon A1 confocal microscope (Nikon Instruments Inc.). To determine % antibody-labelled cells in sections, the total numbers of labelled cells and DAPI+ cells were assessed in Image J using the Cell Counter plugin. To determine the number of labelled cells per µm of basement membrane, the number of labelled cells was divided by the length of the basement membrane.

## Western blotting

Cells were lysed on ice in 1x RIPA buffer (Bio-Rad) supplemented with protease and phosphatase inhibitor cocktails (Roche). Total protein was quantified using a BCA kit (Pierce). Soluble proteins were resolved by SDS-PAGE on 4–15% Mini-PROTEAN TGX gels (Bio-Rad) and transferred onto Trans-Blot 0.2 µm PVDF membranes (Bio-Rad) using the Trans-Blot Turbo transfer system (Bio-Rad). Primary antibody probed blots were visualised with appropriate horseradish peroxidase-coupled secondary antibodies using enhanced chemiluminescence (ECL; Amersham). The ChemiDoc Touch Imaging System (Bio-Rad) was used to image the blots. Quantification of detected bands was performed using Image Lab software (Bio-Rad). Uncropped versions of the western blot presented in that manuscript are available in *Supplementary file 14*.

## Epidermal wholemounts

The procedure was modified from previous reports (*Jensen et al., 1999*). Skin samples from either breast or abdomen were obtained as surgical waste with appropriate ethical approval and treated with Dispase (Corning) overnight on ice at 4°C. The epidermis was peeled off as an intact sheet and immediately fixed in 4% paraformaldehyde for 1 hr. Fixed epidermal sheets were washed and stored in PBS containing 0.2% sodium azide at 4°C. Sheets were permeabilised and stained with specific antibodies in a 24-well tissue culture plate. Image acquisition was performed using a Nikon A1 confocal microscope. 3D maximal projection (1024 × 1024 dpi), volume rendering and deconvolution on stacked images were generated using NIS Elements version 4.00.04 (Nikon Instruments Inc.).

## Antibodies

Antibodies against the following proteins were used: P-ERK (Cell Signaling # 9101; western blot – 1:1000 dilution), ERK (Cell Signaling # 9102; western blot – 1:1000), P-p38 (Cell Signaling # 9211;

western blot – 1:1000), p38 (Cell Signaling # 9212; western blot – 1:1000), Cyclophilin B (R and D # MAB5410; western blot – 1:4000), MKP-3/DUSP6 (Abcam # ab76310; western blot - 1:1000 and R and D Systems # MAB3576-SP; immunostaining – 1:200), PPTC7 (Abcam # ab122548; western-blot 1:250 and Sigma # HPA039335; immunostaining – 1:200), PTPN1/PTP1B (Sigma # HPA012542; western-blot - 1:500; R and D Systems # AF1366-SP; immunostaining – 1:200), PTPN13 (R and D Systems # AF3577; western-blot - 1:300 and immunostaining – 1:200), PPP3CA/Calcineurin A (Sigma # HPA012778; western-blot 1/1000 and R and D Systems # MAB2839-SP; immunostaining – 1:200), DUSP10 (Abcam # 140123; western-blot - 1:1000 and immunostaining – 1:200), TP63 (SCBT # sc-8431 or Abcam # ab735; immunostaining – 1:100), Involucrin (SY5, in-house; immunostaining – 1:500, or DH1, in-house; immunostaining – 1:200), Ki67 (Abcam # ab16667; immunostaining – 1:100) and β1-Integrin (clone P5D2, in-house; immunostaining – 1:200). Species-specific secondary antibodies conjugated to Alexa 488 or Alexa 594 were purchased from Molecular Probes, and HRP-conjugated second antibodies were purchased from Amersham and Jackson ImmunoResearch.

### RNA extraction and RT–qPCR

Total RNA was isolated using the RNeasy kit (Qiagen). Complementary DNA was generated using the QuantiTect Reverse Transcription kit (Qiagen). qPCR analysis of cDNA was performed using qPCR primers (published or designed with Primer3) and Fast SYBR green Master Mix (Life Technologies). RT-qPCR reactions were run on a CFX384 Real-Time System (Bio-Rad). Heatmaps of RT-qPCR data were generated by Multiple Expression Viewer (MeV_4_8) or GraphPad Prism 7. Sequences of qPCR primers are provided in *Supplementary file 13*.

### Statistics

Statistical analysis of the quantifications presented in the Figure legends was performed using GraphPad Prism 7.

## Acknowledgements

This research was funded by the Medical Research Council and the Wellcome Trust. The authors also gratefully acknowledge the use of Core Facilities provided by the financial support from the Department of Health via the National Institute for Health Research (NIHR) comprehensive Biomedical Research Centre award to Guy's and St Thomas' NHS Foundation Trust in partnership with King's College London and King's College Hospital NHS Foundation Trust. Bénédicte Oulès is the recipient of fellowships from Société Francaise de Dermatologie, Collège des Enseignants en Dermatologie de France and Fondation d'Entreprise Groupe Pasteur Mutualité. We thank Giacomo Donati, Arsham Ghahramani and Klaas Mulder for helpful advice and discussions and are grateful to Fredrik Pontén for providing anti-phosphatase antibodies.

## Additional information

#### Competing interests

Fiona M Watt: Deputy editor, *eLife*. The other authors declare that no competing interests exist.

#### Funding

| Funder | Grant reference number | Author |
|--------|------------------------|--------|
| Wellcome Trust | 096540/Z/11/Z | Bénédicte Oulès<br>Kifayathullah Liakath-Ali<br>Gernot Walko<br>Priyalakshmi Viswanathan<br>Fiona M Watt |
| Medical Research Council | G1100073 | Angela Oliveira Pisco |
| Wellcome Trust | 108058/Z/15/Z | Angus I Lamond<br>Tony Ly |

The funders had no role in study design, data collection and interpretation, or the decision to submit the work for publication.

## Author contributions

Ajay Mishra, Conceptualization, Supervision, Validation, Investigation, Visualization, Methodology, Writing—original draft, Writing—review and editing, Performed and analyzed experiments, Performed the statistical analysis; Bénédicte Oulès, Validation, Investigation, Visualization, Writing—original draft, Writing—review and editing, Performed and analyzed experiments; Angela Oliveira Pisco, Conceptualization, Data curation, Software, Formal analysis, Visualization, Methodology, Writing—original draft, Writing—review and editing, Performed the computational analysis for the Boolean networks and the bioinformatic analysis; Tony Ly, Software, Formal analysis, Investigation, Visualization, Methodology, Writing—original draft, Performed and analyzed experiments; Kifayathullah Liakath-Ali, Validation, Investigation, Visualization, Methodology, Writing—original draft, Performed and analyzed experiments; Gernot Walko, Investigation, Methodology, Performed and analyzed experiments; Priyalakshmi Viswanathan, Jagdeesh Nijjher, Investigation, Performed and analyzed experiments; Matthieu Tihy, Software, Developed the automated pipeline for the analysis of the organotypic cultures; Sara-Jane Dunn, Conceptualization, Resources, Software, Formal analysis, Visualization, Writing—review and editing, Performed the computational analysis for the Boolean networks; Angus I Lamond, Conceptualization, Supervision, Funding acquisition, Writing—original draft, Writing—review and editing, Oversaw the experiments; Fiona M Watt, Conceptualization, Supervision, Funding acquisition, Writing—original draft, Project administration, Writing—review and editing, Conceived the project, Oversaw the experiments

## Author ORCIDs

Angela Oliveira Pisco http://orcid.org/0000-0003-0142-2355
Matthieu Tihy http://orcid.org/0000-0002-9314-4657
Angus I Lamond http://orcid.org/0000-0001-6204-6045
Fiona M Watt http://orcid.org/0000-0001-9151-5154

## Decision letter and Author response

Decision letter https://doi.org/10.7554/eLife.27356.035
Author response https://doi.org/10.7554/eLife.27356.036

# Additional files

## Supplementary files

• Supplementary file 1. Log$_2$ fold change of normalised gene expression for all pairwise comparisons of mRNA levels during suspension-induced terminal differentiation. For each condition the mean of n = 3 independent replicates was used and the pairwise fold change comparison is between the means of both samples.
DOI: https://doi.org/10.7554/eLife.27356.015

• Supplementary file 2. Proteomics data for all pairwise comparisons of protein levels at 4, 8 and 12 hr in suspension relative to the 0 hr control. For each condition the mean of n = 3 independent replicates was used and the pairwise fold change comparison is between the means of both samples.
DOI: https://doi.org/10.7554/eLife.27356.016

• Supplementary file 3. Phosphoproteomics data for pairwise comparisons at 4 and 8 hr in suspension relative to the 0 hr control.
DOI: https://doi.org/10.7554/eLife.27356.017

• Supplementary file 4. p-values generated for RT qPCR of TP63 and TGM1 for each conditional time course relative to control time course (siSCR) by two-way ANOVA with Dunnett's multiple comparisons test (related to *Figure 2e,f*).
DOI: https://doi.org/10.7554/eLife.27356.018

• Supplementary file 5. Log2ratio of phosphopeptides over total proteins at 4 hr.
DOI: https://doi.org/10.7554/eLife.27356.019

• Supplementary file 6. Effect of phosphatase knockdown on AP1 transcription factor expression. p-values generated for each conditional time course relative to control time course (SCR) by two-way ANOVA multiple comparisons (for AP1 superfamily factors). p-values generated for RT qPCR of AP1 factors for each conditional time course relative to control time course (siSCR) by two-way ANOVA with Dunnett's multiple comparisons test.

DOI: https://doi.org/10.7554/eLife.27356.020

• Supplementary file 7. Effect of DUSP6 and DUSP10 knockdown on AP1 transcription factor expression. p-values generated for RT qPCR of AP1 factors relative to control cells (siSCR) by two-way ANOVA.

DOI: https://doi.org/10.7554/eLife.27356.021

• Supplementary file 8. Boolean expression patterns and phosphatases interactions used to generate *Figure 4c,d*.

DOI: https://doi.org/10.7554/eLife.27356.022

• Supplementary file 9. p-values generated for RT-qPCR of phosphatases for each conditional time course relative to control time course (siSCR) by two-way ANOVA with Dunnett's multiple comparisons test.

DOI: https://doi.org/10.7554/eLife.27356.023

• Supplementary file 10. One-way non-parametric ANOVA (Friedman test) with Dunn's multiple comparisons test for the effect of overexpressing DUSP6 and DUSP10 on mRNA levels of the pro-commitment phosphatases, determined by RT-qPCR.

DOI: https://doi.org/10.7554/eLife.27356.024

• Supplementary file 11. siRNA library for phosphatase knockdown.

DOI: https://doi.org/10.7554/eLife.27356.025

• Supplementary file 12. shRNA library for phosphatase knockdown.

DOI: https://doi.org/10.7554/eLife.27356.026

• Supplementary file 13. List of qPCR primers.

DOI: https://doi.org/10.7554/eLife.27356.027

• Supplementary file 14. Uncropped versions of the western blots presented in *Figure 3d,g* and *Figure 3 – Figure 4—figure supplement 2c*.

DOI: https://doi.org/10.7554/eLife.27356.028

• Source code 1. Automated measurement of epidermal thickness.

DOI: https://doi.org/10.7554/eLife.27356.029

• Transparent reporting form

DOI: https://doi.org/10.7554/eLife.27356.030

### Major datasets

The following datasets were generated:

| Author(s) | Year | Dataset title | Dataset URL | Database, license, and accessibility information |
|---|---|---|---|---|
| Mishra A, Pisco AO, Watt FM | 2017 | A protein phosphatase network controls the temporal and spatial dynamics of differentiation commitment in human epidermis | https://www.ncbi.nlm.nih.gov/geo/query/acc.cgi?acc=GSE73147 | Publicly available at the NCBI Gene Expression Omnibus (accession no. GSE73147) |
| Mishra A, Pisco AO, Watt FM | 2017 | A protein phosphatase network controls the temporal and spatial dynamics of differentiation commitment in human epidermis | http://proteomecentral.proteomexchange.org/cgi/GetDataset?ID=PXD003281 | Publicly available at ProteomeXchange (accession no. PXD003281) |

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
