## [Decision Letter]

Thank you for submitting your article "A protein phosphatase network controls temporal and spatial dynamics of differentiation commitment in human epidermis" for consideration by *eLife*. Your article has been reviewed by three peer reviewers, and the evaluation has been overseen by a Reviewing Editor and Marianne Bronner as the Senior Editor. The reviewers have opted to remain anonymous.

The reviewers have discussed the reviews with one another and the Reviewing Editor has drafted this decision to help you prepare a revised submission.

Summary:

Commitment of adult stem cells to differentiate is poorly understood. Mishra et al. use human epidermal keratinocytes to explore this process. The authors induce keratinocyte differentiation using suspension cultures and perform analysis of gene and protein expression. This analysis revealed a set of phosphatases that promote keratinocyte differentiation, while another phosphatase, DUSP10, antagonizes commitment to differentiate. The authors validate their findings using siRNA knockdown and in vitro epidermal reconstitution assays. Pro-commitment phosphatases negatively regulate MAPK signaling and positively regulate the transcription factor AP1 to promote differentiation. In silico analysis suggests that commitment to differentiate is controlled by an autoregulatory network of phosphatases in human epidermal keratinocytes.

The manuscript is well written and the data shown are of general interest to the field of cell biology. The authors have suggested the following revisions:

Essential revisions:

1) The reviewers requested clarification of the data such as the mathematical modeling. In particular, the mathematical modeling (Boolean networks) suggests temporal regulation. This concept is not explained clearly, no model, formula, or conditions are specified. It is difficult to understand why the data support the conclusions.

2) Additionally, the characterization of the temporal regulation of the network is unclear.

3) Knockdown the genes permanently (shRNA delivery by lenti or retroviruses) or knockout of DUSP6, PTPN1, and PPP3CA. This is particularly important in being able judge the phenotypes that they have. For example, knockdown of the above mentioned factors didn't prevent reconstituted epidermal tissue from differentiating but resulted in the presence of intermediate state cells with both basal and differentiated markers. These skin constructs are taken out for 3 weeks which is outside the range of the efficacy of siRNAs (3-5 days), it is unclear how to interpret the results. Along these lines, DUSP10 knockdown should be compared to scrambled siRNA rather than DUSP6 knockdown (DUSP6 knockdown reduced expression of AP1 transcription factors and thus comparing anything against this group could make the DUSP10 knockdown look like its artificially increasing the AP1 transcription factors.)

4) The Western blot image is not representative or is at a low resolution. Loading controls are inconsistent (see Figure 3 scrambled and siDUSP6). According to the quantification, P-ERK levels are the highest after PTPN13 loss. In the image, pERK levels seem to be higher in PPTC7.

5) Statistics: Throughout the manuscript, statistics are needed. In particular, in Figure 2 and Figure 3.

---

## [Author Response]

Essential revisions:1) The reviewers requested clarification of the data such as the mathematical modeling. In particular, the mathematical modeling (Boolean networks) suggests temporal regulation. This concept is not explained clearly, no model, formula, or conditions are specified. It is difficult to understand why the data support the conclusions.

To address the reviewers’ concerns we have clarified the description of the Boolean network analysis in the main text, cited additional references and expanded the Materials and methods section to provide more information about how the modelling was performed. We now clearly refer to Yordanov et al. (2016), where all mathematical details of the modelling approach are explained. We also provide the necessary information to enable readers to understand how we implemented this approach, including the set of possible and definite interactions that comprise our Abstract Boolean Networks (ABNs) (Figure 4, newly included Figure 4—figure supplement 7, and revised Figure 4—tablesupplements 8, 9) and the definition of the constraints that we encoded into RE:IN / RE:SIN. Together with the expanded Materials and methods section, the revised manuscript provides a comprehensive explanation of the modelling approach we implemented.

2) Additionally, the characterization of the temporal regulation of the network is unclear.

We have revised the manuscript to make this clear. First, we tested whether a single Boolean network could be consistent with the experimentally observed changes in gene expression states associated with different lengths of time in suspension. Briefly, we exploited the concept of an Abstract Boolean Network (ABN) to define allpossible networks that could exist between the phosphatases we considered. Thus, we encoded an ABN in which possible positive and negative interactions existed between each pair of phosphatases. We then used the gene expression data to define expected states along the differentiation trajectory, which should be satisfied by a valid network model. We used the time measurements (Figure 1) as a proxy for the number of steps along the trajectory (e.g. the expression measurements at 3h should occur after 3 steps on the network trajectory). Finally, RE:IN allowed us to test all concrete networks defined by this ABN. We found that, computationally, a single network of positive and negative interactions between the protein phosphatases (Figure 4—figure supplement 7A) could not simulate the differences in gene expression observed experimentally as a function of time in suspension (Figure 2). We therefore introduced a new concept: the interaction network between the phosphatases could change over time. We defined ABNs for each time step (the topologies of which were informed by our knockdown experiments) and sought to identify networks for each time point that were consistent with the measured gene expression states. We translated our experimental data, relating both to changes in time and the effects of PKC inhibition and TSA treatment, into computational steps (‘active’ for upregulated, ‘inactive’ for downregulated), as shown in Figure 4. In this way every iteration step of the network analysis corresponds to temporal progression in real time. With this approach we could successfully model the changes in the phosphatase interaction network that occur over time in suspension. The revised manuscript now includes an extensive explanation of the iteration scheme used to perform the computational analysis (Materials and methods subsection “Dynamical Boolean Network Analysis”; Results subsection “A protein phosphatase interaction network acts as a switch to transition cells between the stem and differentiated cell compartments” main text pages 7 – 10).

3) Knockdown the genes permanently (shRNA delivery by lenti or retroviruses) or knockout of DUSP6, PTPN1, and PPP3CA. This is particularly important in being able judge the phenotypes that they have. For example, knockdown of the above mentioned factors didn't prevent reconstituted epidermal tissue from differentiating but resulted in the presence of intermediate state cells with both basal and differentiated markers. These skin constructs are taken out for 3 weeks which is outside the range of the efficacy of siRNAs (3-5 days), it is unclear how to interpret the results.

We have now repeated the epidermal reconstitution experiments using lentiviral shRNA vectors (Figure 2—figure supplement 4) and observe similar effects to those of the siRNAs. Although the efficacy of the siRNAs is indeed a matter of days, the consequences of knocking down the phosphatases in stem cells are nevertheless long-term since they affect cell fate decisions that are manifested during the terminal differentiation programme.

Along these lines, DUSP10 knockdown should be compared to scrambled siRNA rather than DUSP6 knockdown (DUSP6 knockdown reduced expression of AP1 transcription factors and thus comparing anything against this group could make the DUSP10 knockdown look like its artificially increasing the AP1 transcription factors.)

We believe that the reviewers are referring to Figure 3. In these experiments the values for siDUSP6 and siDUSP10 are indeed expressed relative to the siSCR control. This point is now clarified in the Figure legend.

4) The Western blot image is not representative or is at a low resolution. Loading controls are inconsistent (see Figure 3 scrambled and siDUSP6). According to the quantification, P-ERK levels are the highest after PTPN13 loss. In the image, pERK levels seem to be higher in PPTC7.

We have replaced this panel with a new panel in which the quality of the Western blots has been improved. We have used a different loading control (Cyclophilin B) and included quantitation for both phospho-ERK and total ERK (Figure 3).

5) Statistics: Throughout the manuscript, statistics are needed. In particular, in Figure 2 and Figure 3.

These have now been included.